# Resolution and Field of View Invariant Generative Modelling with Latent Diffusion Models

**Ashay Patel** [1]                     ASHAY.PATEL@KCL.AC.UK
**Mark S. Graham** [1]                   MARK.GRAHAM@KCL.AC.UK
**Vicky Goh** [1]                      VICKY.GOH@KCL.AC.UK
**Sebastien Ourselin** [1]               SEBASTIEN.OURSELIN@KCL.AC.UK
**M. Jorge Cardoso** [1]                M.JORGE.CARDOSO@KCL.AC.UK
[1] *Department of Biomedical Engineering, School of Biomedical Engineering & Imaging Sciences, King's College London, London, UK*

**Editors:** Accepted for publication at MIDL 2024

## Abstract

Large dataset requirements for deep learning methods can pose a challenge in the medical field, where datasets tend to be relatively small. Synthetic data can provide a suitable solution to this problem, when complemented with real data. However current generative methods normally require all data to be of the same resolution and, ideally, aligned to an atlas. This not only creates more stringent restrictions on the training data but also limits what data can be used for inference. To overcome this our work proposes a latent diffusion model that is able to control sample geometries by varying their resolution, field of view, and orientation. We demonstrate this work on whole body CT data, using a spatial conditioning mechanism. We showcase how our model provides samples as good as an ordinary latent diffusion model trained fully on whole body single resolution data. This is in addition to the benefit of further control over resolution, field of view, orientation, and even the emergent behaviour of super-resolution. We found that our model could create realistic images across the varying tasks showcasing the potential of this application.

**Keywords:** Generative Modelling, Latent Diffusion Models, Multi-Resolution, Field of View Invariant

## 1. Introduction

Deep learning methods have achieved remarkable growth in recent years in both natural imaging and medical imaging analysis. Nevertheless deep learning methods are data-hungry, and the size of publicly available medical datasets are much smaller than their natural image counterparts. These sizes can be drastically smaller depending on the area of focus, with the average dataset being less than 200 samples (Kiryati and Landau, 2021). This limitation is also increased by the stringent data requirements for medical imaging. Most work requires data to be of the same resolution aligned to an atlas, in addition to showcasing the same field of view. This can consequently require large amounts of preprocessing and/or simplified model design which can result in poor generalisation (Varoquaux and Cheplygina, 2022; Decuyper et al., 2021; Dinsdale et al., 2022). A restriction we hope to overcome with our proposed model invariant to such characteristics, as displayed in figure 1.

Being able to produce high quality synthetic data can provide the medical imaging community a promising alternative to improve ones ability to carry out the appropriate

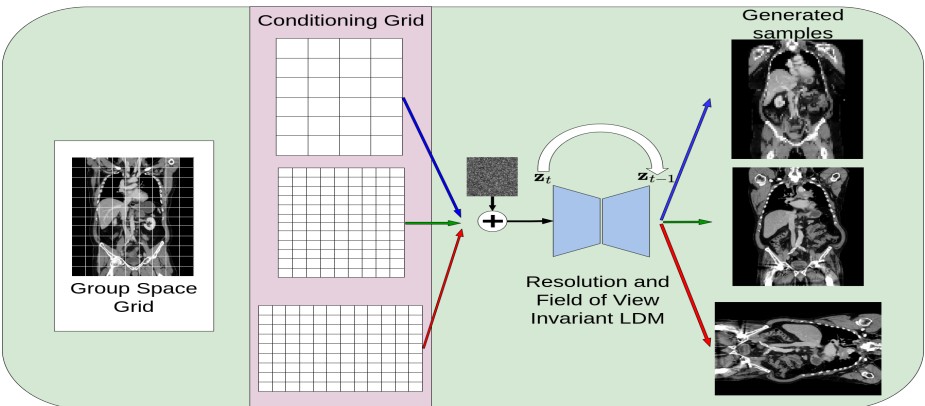

Figure 1: Spatial Conditioning provided by generating any arbitrary affine transform grid that showcases the mapping from a group space grid to a new space that defines the images orientation, resolution and field of view. The coordinate grid is then concatenated to the input fed into the diffusion model to control sampling. The three paths show a coarse affine coordinate grid to generate a low resolution image (blue), a fine grained grid for high resolution image (green) and a rotated fine grained image to create a rotated high resolution image (red).

research on larger scales (Jordon et al., 2020, 2022; Wang et al., 2021). The use of generative models have shown a promising solution for this, by learning the probability density function of the underlying training data, which they can then use to create new realistic examples.

Variational Autoencoders (VAEs) (Baur et al., 2020) are a simple baseline for generative modelling in imaging, however they have been shown to produce blurry samples with a number of better alternatives. Until recently, Generative Adversarial Networks (GANs) have been at the forefront of generative modelling research (Creswell et al., 2017; Kwon et al., 2019; Wang et al., 2020). However an added difficulty within medical imaging, especially with 3D images, is the computational requirements. Further limitations of such networks are commonly documented drawbacks, namely unstable training, mode collapse and failure to converge (Kodali et al., 2017). In addition they are hard to guide via conditioning.

Recently diffusion models have shown great promise and development showing state-of-the-art results for generating synthetic images, surpassing GANs in sample quality and diversity (Dhariwal and Nichol, 2021). They have also been shown to effectively create images based on various conditionings i.e. class-based, image and text-based (using techniques like classifier-free guidance (Ho and Salimans, 2022)). Notable real-life examples include Latent Diffusion Models (LDMs) (Rombach et al., 2021) and DALL-E (Ramesh et al., 2021). Latent diffusion models specifically have shown promise in medical imaging as they make use of an Autoencoder model to compress data, and then a diffusion model to work on the compressed latent space. The work in (Pinaya et al., 2022) and (Graham et al., 2023) showcases the use of LDMs in medical imaging for generative modelling and out-of-distribution detection respectively displaying state-of-the-art results.

Even so, current methods in generative modelling lack in diversity of image geometry. In medical imaging, they are all designed to generate images at a particular dimension, resolution and field-of-view. Models able to train and infer on varying data types be it resolution and field-of-view is still a relatively new concept, however as demonstrated in (Patel et al.,

2023), they can have a number of benefits including maximising data utilisation, reducing data preprocessing and improving generalisability. As such being able to compliment such a model and producing a generative model with the added flexibility of being able to control the resolution, field-of-view and orientation can have significant benefits to further pursuing the goal of a singular model with performance capabilities not limited to single group space. To do so we should be modelling both the image and the affine mapping to a group space.

In this study we aim to showcase the use of a Latent Diffusion Model (LDM) with a spatial conditioning that defines both the affine transform grid to a group space and voxel sizing of an image that allows us to control the resolution of images generated, in additional to the field-of-view and orientation. We demonstrate this work on a dataset of 1014 3D whole body CT scans. We showcase that the addition of this spatial conditioning still generates samples with comparable quality to an LDM trained only on whole body images of one resolution, but with the added benefit of control over the generation. Additionally, without specifically training for such a task, we show a unique emergent behaviour by this model for the downstream task of image super-resolution to any specified resolution.

## 2. Method

### 2.1. Generative Modelling

Our methodology is based around LDMs, which use an autoencoder-based model to compress input data into a lower-dimensional latent representation in conjunction with the generative modelling properties of a diffusion model. As we are working with high-dimensional 3D data, the compression model is an essential step. For this we trained a Vector Quantized-Variational Autoencoder (van den Oord et al., 2017).

**Vector Quantized-Variational Autoencoder:** The VQ-VAE consists of an encoder part that maps an image $\mathbf{x} \in \mathbb{R}^{H \times W \times D}$ onto a compressed latent representation $\mathbf{z} \in \mathbb{R}^{h \times w \times d \times n_z}$ where $n_z$ is the latent embedding vector dimension. The compressed latent representation $\mathbf{z}$ is then passed through a quantization block where each feature column vector is mapped to its nearest codebook vector via a euclidean distance metric. Each spatial code $\mathbf{z}_{ijl} \in \mathbb{R}^{n_z}$ is then replaced by its nearest codebook element $e_k \in \mathbb{R}^{n_z}, k \in 1, ..., K$ where $K$ denotes the codebook vocabulary size, resulting in the quantized latent space $\mathbf{z}_q$. Given $\mathbf{z}_q$, the VQ-VAE decoder then reconstructs the observations $\hat{\mathbf{x}} \in \mathbb{R}^{H \times W \times D}$. In practice we employ a VQ-GAN model (Esser et al., 2020), to produce higher quality reconstructions by employing adversarial training. See appendix B for implementation details.

**Denoising Diffusion Probabilistic Models:** Once we have trained the VQ-GAN, we can then train a Denoising Diffusion Probablistic Model (DDPM) (Ho et al., 2020) on the sample latents (de-quantized latents). During training, noise is added to the latent $\mathbf{z}$ with respect to a timestep $t$ using a fixed Gaussian noise schedule defined by $\beta_t$ rendering a noised sample $\mathbf{z}_t$ such that:

$$q(\mathbf{z}_t|\mathbf{z}_0) = \mathcal{N}\left(\mathbf{z}_t \sqrt{\bar{\alpha}_t}\mathbf{z}_0, (1 - \bar{\alpha}_t)\mathbf{I}\right) \tag{1}$$

where $\mathbf{z}_0$ refers to a noise free latent $\mathbf{z}$ and $0 < t \leq T$. Additionally $\alpha_t := 1 - \beta_t$ and $\bar{\alpha}_t := \prod_{s=1}^t \alpha_s$. The aim is to generate a network that can perform the reverse process or denoising process which can be presented as the Gaussian transition:

$$p_\theta\left(\mathbf{z}_{t-1} \mid \mathbf{z}_t\right) = \mathcal{N}\left(\mathbf{z}_{t-1} \mid \boldsymbol{\mu}_\theta\left(\mathbf{z}_t, t\right), \boldsymbol{\Sigma}_\theta\left(\mathbf{z}_t, t\right)\right) \tag{2}$$

From (Ho et al., 2020) we can train our network $\epsilon_\theta(\mathbf{z}_t, t)$ to predict the noise directly in the forward noising process, $\epsilon$. We can then train with the simplified objective loss $L(\theta) = \mathbb{E}_{t,\mathbf{z}_0,\epsilon} \left[ \|\epsilon - \epsilon_\theta(\mathbf{z}_t)\|^2 \right]$ and denoise according to :

$$\mathbf{z}_{t-1} = \frac{1}{\sqrt{\alpha_t}} \left( \mathbf{z}_t - \frac{\beta_t}{\sqrt{1 - \bar{\alpha}_t}} \epsilon_\theta(\mathbf{z}_t, t) \right) + \sigma_t \mathbf{n} \tag{3}$$

where $\mathbf{n} \sim \mathcal{N}(0, \mathbf{I})$ and $\sigma_t^2 = \beta_t$

For training, we use a noise schedule of $T = 1000$ with a scaled linear variance noise schedule with $\beta_0 = 0.0015$ and $\beta_T = 0.0195$. See appendix C for implementation details.

## 2.2. Spatial Conditioning

As the VQ-GAN and DDPM are purely convolution, we are able to input data of varying dimension i.e. allowing for data at varying resolution and FOVs. However this requires a form of spatial conditioning that enables modelling of such geometric variations. To produce images of varying resolution, FOV and orientation we need to model both the image and the geometry of said image represented by the affine mapping of the given image to a group space plus the grid size (or voxel dimensions). This affine transformation gives us an explicit coordinate frame to work from to control the image generation.

**VQ-VAE Spatial Conditioning:** To control the generation we require a form of conditioning to help the VQ-VAE encode and reconstruct data at varying resolutions and also to help steer the denoising process in the DDPM. For this we take inspiration from the work in CoordConv (Liu et al., 2018). This is further adapted to model more intricate image geometries to essentially define any affine transform from one image to a group space, as similarly explored in (Patel et al., 2023) for anomaly detection. This accounts for some level of spatial awareness in the data fed into the models. A CoordConv layer is a concatenation of channels to the input image referencing a predefined coordinate system. After concatenation, the input can then be fed through any given convolutional layer as normal. For a 3D image, we would thus have 3 coordinates, $ijk$, where the $i$ coordinate channel is a $h \times w \times d$ rank-1 matrix with its first row filled with 0's, second row with 1's and so on. This would then of course be the same for the j coordinate channel, but with columns filled with constant values not rows, and likewise for the k channel in a depth-wise fashion. These channels would then be normalised between [0,1]. The main utilisation of the coordinate system in this work however is due to the 0-1 scale across the channels regardless of image resolution. For example 2 whole body images of varied resolution and dimensions would still have coordinate channels ranging from 0-1 along each axis, but with differing increments between each voxel, thus conveying the notion of spatial resolution i.e. showcasing the grid sizing of the affine. It was found in (Patel et al., 2023), that training a VQ-VAE on images with large variations in resolution resulted in poor reconstructions, and the coordinate channels helped to overcome this issue. Furthermore, we can further adapt these coordinates to convey a notion of the field-of-view (FOV) present in the image. It is common when working with various types of scans of the body, that a whole-body scan is not always taken. This data can still be useful if we can provide a way for our models to work with data of varying FOVs. We can convey this by adapting the 0-1 range of the coordinate channels. In our case a whole body image (upper leg to neck in this work)

represents our full range of [0,1], where in the axial plane 0 represents the bottom of the body and 1 represents the top. We then contract the range to represent a different FOV, i.e. perhaps [0.5,1] to showcase a scan of the upper body only. A graphic demonstration of this can be found in Figure 2 also highlighting changes to the coordinate grid as a result of a rotation. Note that while the proposed model assumes translation, scale and rotational changes, as we are modelling the affine transform to a group space, this can be trivially extended to a full affine mapping of the coordinate system including shearing.

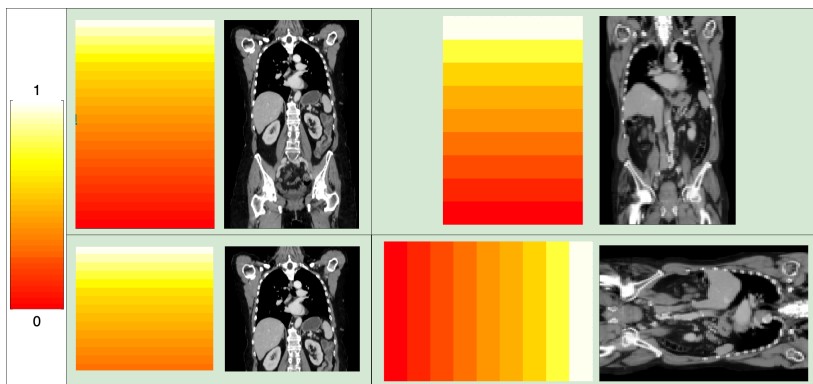

Figure 2: Coordinate System for the axial channel displaying an example affine grid for whole-body images at 2 different resolutions (top row), cropped (bottom left) and rotated (bottom right)

**LDM Spatial Conditioning:** To include the spatial conditioning in the DDPM, the coordinate channels used in the VQ-GAN encoding, are downsampled to the size of the latent space via an average pooling with kernel size and stride of 8 , equaivalent to the downsampling applied in the VQ-GAN. This then gives us 3 spatial channels at the latent dimension that define the affine and grid sizing in order to steer the generation process. These channels are then concatenated as channels to the input to condition the generation. For sampling, we define the extremes of the coordinate channels and desired dimension, for which we can then linearly interpolate the intermediate values to give the conditioning.

### 2.3. Emergent Behaviour – Super-resolution

Further to the generative modelling capabilities of our model and as a result of our model being able to control generation at varying resolutions, we showcase an emergent behaviour of our model - super-resolution. This requires no further training and is purely an emergent behaviour of the training mechanism and conditioning employed in this methodology. As reported in the work of (Ho et al., 2020), different $t$ values are responsible for modelling different image features, with higher values associated with coarser features, and lower $t$ values responsible for finer details. We can take this information to define a suitable denoising schedule to super-resolve an image. The method works by taking a low resolution image, and upscaling it via a linear upscaling to the desired dimension we require. We then encode this to get our latent space. At which point we partially noise the encoded image and then proceed to denoise the image. Utilising the information that finer grained detailed occur at lower t-values, and that the larger scale details are already present from the original upscaled image we use a low $t$ value of $t = 100$ chosen empirically from visualising samples

qualitatively. It was found that higher $t$ values eventually resulted in larger changes in the structure of the CT image away from the original sample, although sample quality was higher. To overcome this we repeated the noising and denoising process. The advantage of this was to slowly improve the image resolution over cycles without deviating from the original CT content. The final process chosen made use of 5 cycles of noising the image to $t= 100$ and denoising. For this we use a Denoising Diffusion Implicit Model Scheduler which does not add any noise during each denoising step to prevent the image moving away from the original sample. One highlight of this emergent behaviour is that we did not require any paired low and high resolution images, and additionally we are able to super-resolve from any arbitrary resolution to any higher resolution.

## 3. Results

### 3.1. Experiment 1 - Generative Modelling

Table 1: Quantitative Results showcasing FID score for varying sample types

| Experiment | Baseline LDM | Spatial Conditioned LDM |
| --- | --- | --- |
| Whole Body High Resolution | 0.0378 ±0.0032 | **0.0302** ±0.0036 |
| Whole Body Medium Resolution | 0.0406 ±0.0047 | 0.0414 ±0.0045 |
| Whole Body Low Resolution | 0.0712 ±0.0059 | **0.0562** ±0.0046 |
| Upper Body Medium Resolution | 0.0586 ±0.0059 | 0.0594 ±0.0063 |
| Half Body Medium Resolution | 0.0451 ±0.0052 | 0.0472 ±0.0041 |
| Whole Body Rotated | 0.0442 ±0.0048 | 0.0491 ±0.0057 |

Our first experiment is designed to demonstrate the generative modelling capabilities of our method for both whole-body CT images at a fixed intermediate resolution in addition to being able to generate images at varying resolutions, FOVs and orientation. Prior research has showcased that LDMs display state-of-the-art performance for generative modelling, and as such we will compare our results to that of an LDM identical in architecture to our proposed method without any spatial conditioning, trained solely on whole body CT images at a fixed resolution ($192 \times 192 \times 208$). To compare to our geometry invariant model, we will take whole-body samples from the baseline LDM and apply transforms to generate the desired sample i.e. via scaling, cropping and rotating. We aim to showcase that our model showcases no performance degradation compared to a whole-body fixed resolution model. This is complimented with the added benefit of being able to control sample attributes without any reduction in sample quality. We test our model at 3 resolutions ($96 \times 96 \times 104$, $192 \times 192 \times 208$ and $272 \times 272 \times 288$), in addition to 2 partial body views and a rotated orientation. To obtain quantitative metrics we use the Fréchet Inception Distance (FID) (Heusel et al., 2017) to measure how realistic the synthetic images are, with smaller values indicating images are more similar to the distribution of the real images. To calculate these values, we followed a similar approach to (Sun et al., 2022), by extracting features using a pretrained Med3D (Chen et al., 2019). For each experiment 1000 samples were generated to calculate the FID scores. Statistical tests were carried out via bootstrapping, and statistically significant findings ($p < .01$) are reported in bold in Table 1.

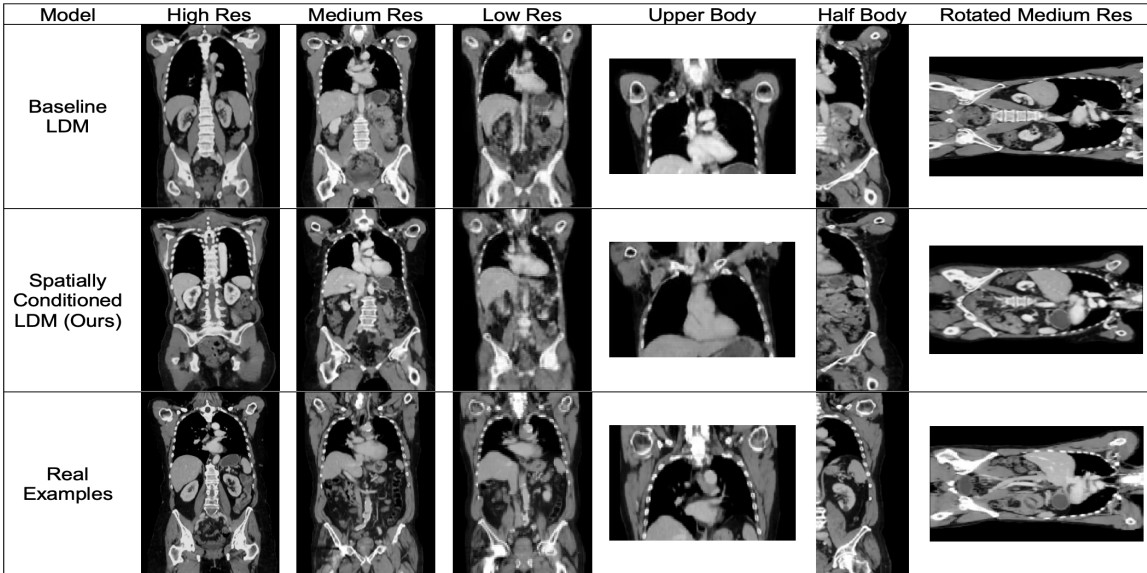

Figure 3: Qualitative results of samples generated by the Baseline LDM, Spatially Conditioned LDM and ground truth examples from training data

From table 1 we can generally see in cases where any scaling is required i.e. for the low and high resolution whole body experiments, our model shows superior performance, highlighting our model's ability to generate more realistic images and avoid any inaccuracies associated with resampling. With regards to the final three experiments that are simple transforms of rotating or cropping, we see roughly equal performance between both models, with our spatially conditioning LDM falling slightly short, albeit a statistically insignificant difference. It is not expected that our model would outperform in this case due to the non-intrusive transformations applied to the baseline, however this does showcase that under these conditions, our model generates consistent and structurally sound samples with almost no degradation in performance.

### 3.2. Experiment 2 - Super-resolution

Our second experiment showcases the emergent behaviour of our model for super-resolution without further training. For this we showcase low resolution samples and super-resolve them to 2 differing higher resolutions. For each sample we generate the super-resolved image 50 times and calculate the mean absolute error (MAE) between the original high resolution image and the super resolved images, in addition to the standard deviation across samples.

We can see from Figure 4 that our model performs well to super-resolve the images to different resolutions, with limited error. Generally most of the error can be seen around the edges of large and higher intensity structures i.e. bone structures etc where finer detail is lost in the low resolution image. The standard deviation maps also highlight areas of high variation over each reconstruction. These maps are telling, as we can see from Figure 4 that there is a high correlation between the standard deviation and the mean error. This shows us that although there is some error, the model understands that it is uncertain around these areas showcased by the correlated high standard deviation and error.

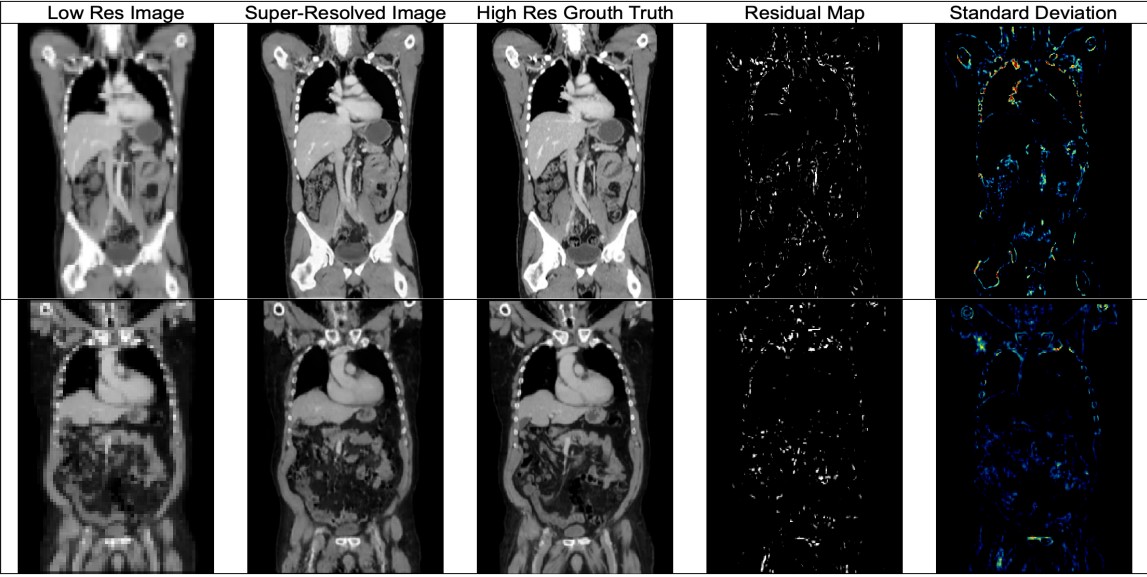

| Low Res Image | Super-Resolved Image | High Res Grouth Truth | Residual Map | Standard Deviation |

Figure 4: Example samples super-resolved using LDM. Columns from left to right showcase: low resolution image (1st), super-resolved image output (2nd), grounth truth high resolution image (3rd), MAE map across reconstructions (4th), standard deviation map across reconstructions (5th). Top sample is super-resolved from $104 \times 104 \times 112$ to $272 \times 272 \times 288$ and the bottom is super-resolved from $88 \times 88 \times 96$ to $192 \times 192 \times 208$ in voxel dimensions.

## 4. Conclusion

In our study, we have shown to effectively train a latent diffusion model to generative synthetic CT data at varying resolutions, fields of view and orientation. We showcase that our model is equal in performance to that of a model trained only on whole body data, with the necessary transforms applied post generation to produce a sample of given resolution or field of view and orientation. Even so in cases where scaling would be required, our model shows improved FID scores, showing greater consistency in the samples against inaccuracies brought about by resampling. Furthermore we show as an emergent behaviour of our training scheme, our model can be used to super-resolve images from any arbitrary lower resolution to any higher resolution, whilst also maintaining some awareness of areas of uncertainty and inaccuracies in the super-resolved images. This work brings to light the efficacy and potential of modelling not only the image, but the image geometry, represented by the affine to a group space with respective grid sizing to enable a better interaction with varying image characteristics. We hope this works warrants further efforts into such developments for a wider scope of applications in medical imaging.

## Acknowledgments

This research was supported by Wellcome/ EPSRC Centre for Medical Engineering (WT203148/Z/16/Z), Wellcome Flagship Programme (WT213038/Z/18/Z), The London AI Centre for Value-based Heathcare and GE Healthcare.

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

## Appendix A. Data

The Data used for this work makes use of AutoPET (Gatidis et al., 2022; Clark et al., 2013) (1014 scans). This data contains paired PET/CT whole body data for which we use the CT data. The original voxel dimensions of this data are very high resolution at $0.8 \times 0.8 \times 2.5mm$. However to cater to memory restrictions and increase computational times and costs we decided to resample all the data to $1.6 \times 1.6 \times 2.5mm$ dimensions. Additionally we clipped all images to obtain the soft-tissue window of the CT images, although not a necessary step, we found when visually and qualitatively assessing the images, this window made it easier to assess image quality and would be more appropriate to demonstrate our results. From the whole-body data, all images then had the full coordinate channels added to them ranging from 0-1 across each dimension due to the whole body nature. Then during training of both the VQ-VAE and DDPM, both random crops, random resizing and rotations. The range of resolutions seen were such that for a whole body image the dimensions ranged from $272 \times 272 \times 288$ at the highest resolution to $96 \times 96 \times 104$ at the lowest resolution.

## Appendix B. VQ-GAN Implementation

The VQ-VAE model was trained using a jukebox loss that is given as:

$$L_{VQVAE} = \|(\mathbf{x} - \hat{\mathbf{x}})\|_2^2 + \||STFT(\mathbf{x})| - |STFT(\hat{\mathbf{x}})|\|_2^2 + \beta \|z_e(\mathbf{x}) - sg[\mathbf{e}]\|_2^2 + \|sg[z_e(\mathbf{x})] - \mathbf{e}\|_2^2 \quad (4)$$

where $sg$ stands for a stop gradient operator to stop gradients from flowing back into their argument. The loss used from (Dhariwal et al., 2020) uses a spectral loss component that is based on the magnitude of the Fourier Transformer of the original and reconstructed

image. From equation 4 the first term seen is the L2 pixel loss, whilst the second term represents the spectral loss between the original and reconstruction. Here SFTF stands for the short-time Fourier transform. The third term is the commitment cost to ensure the encoder commits to the codebook. The final term is to move the codebook embedding vectors towards the encoder output. We replace this final term with an exponential moving average update for the codebook as implemented in (van den Oord et al., 2017). During training, a $\beta$ of 0.25 was used.

To adjust the network to a VQ-GAN we additionally apply adversarial training with the existing VQ-VAE architecture and losses. In doing so the loss function is further amended to include a perceptual loss (Takaki et al., 2018) that helps to preserve spatial consistency, making use of the lpips library (Zhang et al., 2018). This loss is randomly applied to 50% of slices across each plane.

The architecture used for this work in the VQ-VAE makes use of an encoder with three downsampling layers containing convolutions with stride 2 and kernels of size 4, each followed by a ReLU activation and 3 residual blocks. Each residual block is made up of convolutions with kernel size 3, followed by a ReLU, another convolution with kernel size 1 and another ReLU. The decoder architecture mirrors that of the encoder except that it uses transposed convolutional layers with stride 2 and kernel size 4 for upscaling of the image. This work made use of a VQ-VAE codebook with 512 atomic elements (vocabulary size), each of length 8. An ADAM optimiser was used with a learning rate of 1e-4 and an exponential learning rate decay with a gamma of 0.9999. Training was run for 600 epochs with a batch size of 3.

## Appendix C. DDPM Implementation

The DDPM was trained following the objectives described in Section 2.1. For the baseline and the spatially conditioned LDM the model consisted of 3 layers of downsampling with 128, 256 and 256 channels at each level respectively. Furthermore self-attention is applied at the bottleneck of the network, i.e. after 3 layers of downsampling, applied with 8 attention heads .

The models were trained for 6000 epochs with a learning rate of $2.5e-5$ with an ADAM optimiser. For the baseline input data underwent the following transforms: gaussian noise, contrast adjustment, intensity shifts, translations, and elastic deformations. Similarly for the proposed spatially conditioned model, in addition to these transforms, random crops, resizing and rotations were used. For the range of resolutions seen during training, an normally oriented whole body image had dimensions range from $272 \times 272 \times 288$ at the highest resolution to $96 \times 96 \times 104$ at the lowest resolution. The codebase used for the DDPM model makes use of the Monai Generative framework (Pinaya et al., 2023).

