# OpenReview forum: "Resolution and Field of View Invariant Generative Modelling with Latent Diffusion Models"
_MIDL.io/2024/Conference — MIDL 2024 Poster_

### Official Review · Reviewer_wAAw · 2024-02-28

**Confidence:** 3
**Preliminary Rating:** 4
**Recommendation:** Poster
**Final Rating:** 4

**Summary:**

The authors use a diffusion model to generate more similar images with various properties (resolution, etc...) for improving the training of algorithms to be subsequently applied. They also show that this tool can naturally achieve the creation of superresolution images, which may also be of interest in this field.

**Strengths:**

The paper is well written, with a clear exposition of the problem, method and results. The produced images seem to be of good quality, preserving important features of the original dataset better than previous methods.

**Weaknesses:**

The only few weaknesses I would find are that a single dataset was used to assess the quality of the produced images, which may not be sufficient for a full-proof test of this algorithm, if one wants to use it on other types of images. There is also no example on how such images could be used later, and if they provide any added values.

**Detailed Comments:**

The diagram in figure 1 could be improved a bit in terms of readability, some arrows/text are rendered very small, but there is space in the figure.

**Justification Of Final Rating:**

Very good, I do not have any further comments.                                                                                                                                                            .

**Justification Of The Preliminary Rating:**

I am not an expert on diffusion models, but this work seems of reasonable quality to be discussed. I would have liked a more thorough testing of this algorithm on other datasets, or comparison with previous methods, and an example of the added value of having produced more images. I would guess if they are too similar to the original dataset, the added value is limited, but if they are too different they will 'blur' the dataset and reduce the quality of later analysis.

**Questions To Address In The Rebuttal:**

I don't have any major questions to be addressed.

**Special Issue:**

No

---

> ### Author Response · Authors · 2024-03-16
>
> Q1: The diagram in figure 1 could be improved a bit in terms of readability, some arrows/text are rendered very small, but there is space in the figure.
>
> A1: Thank you for your recommendation on the figure. We would of course want to ensure maximum readability and have amended figure 1 based off of your recommendations.
>
>
> Q2: I am not an expert on diffusion models, but this work seems of reasonable quality to be discussed. I would have liked a more thorough testing of this algorithm on other datasets, or comparison with previous methods, and an example of the added value of having produced more images. I would guess if they are too similar to the original dataset, the added value is limited, but if they are too different they will 'blur' the dataset and reduce the quality of later analysis.
>
>
> A2: We agree that further testing of this work on further datasets would have been useful in the validation, however the use of diffusion models in generative modelling on a range of modalities is common in the literature and has been shown to work, but we are equally sure that the suggested spatial conditioning would generalise to various modalities. As this paper focused on a new methodology, not validation we are of course limited by the page restrictions to showcase a wide evaluation whilst also communicating the method. However, we are pushing our future work to better evaluate these methods and showcase their utility on various datasets from varying modalities and dataset types. This would also include evaluation of further downstream tasks utilising the generated data. Nevertheless there are a number of works [1,2] that showcase the use of synthetic data to aid training for further downstream tasks that when complimented with real data improve performance, as such we are sure that this work can be beneficial to enhance the performance of other downstream tasks.
>
> [1] - Fernandez, V., Pinaya, W.H.L., Borges, P., Tudosiu, P.D., Graham, M.S., Vercauteren, T. and Cardoso, M.J., 2022, September. Can segmentation models be trained with fully synthetically generated data?. In International Workshop on Simulation and Synthesis in Medical Imaging (pp. 79-90). Cham: Springer International Publishing.
> [2] - Llugiqi, M. and Mayer, R., 2022, August. An empirical analysis of synthetic-data-based anomaly detection. In International Cross-Domain Conference for Machine Learning and Knowledge Extraction (pp. 306-327). Cham: Springer International Publishing.

---

### Official Review · Reviewer_uASK · 2024-02-28

**Confidence:** 3
**Preliminary Rating:** 3
**Recommendation:** Poster
**Final Rating:** 3.5

**Summary:**

This paper presents a latent diffusion model that is invariant to the resolution and field of view of the data. The method consists of a VQ-GAN to obtain the sample latents and a DDPM based on these latents, both enriched with spatial encoding. The model is trained on whole-body CT images modified with different resolutions, FOVs, and orientation, demonstrating an improved generation quality as measured by FID in comparison to a baseline LDM without the spatial conditioning. An emergent super-resolution capability is also discussed and demonstrated.

**Strengths:**

The proposed spatial conditioning to the LDM is interesting and addresses a real-world adoption gap for these models.

The experimental results provide promising evidence for the improvements the proposed spatial conditioning achieved compared to the baseline LDM.

The emergent behavior of the model in enabling super-resolution is interesting.

**Weaknesses:**

The experiments considered only one dataset with a limited variety of FOV and orientations to demonstrate the real-world applicability of the method presented.

The presented quantitative results miss statistics from multiple seeds of the model training results.

For image super resolution, it is not clear if it requires all the resolutions to be present in the training data?

**Detailed Comments:**

Please see above.

**Justification Of Final Rating:**

The authors addressed my question on adding statistics to the results. My two other main comments were addressed as future work, although given the nature of the proof-of-concept I think the paper will be a good addition to MIDL.

**Justification Of The Preliminary Rating:**

This paper presents an interesting approach (spatial conditioning) to enable LDM to be invariant to image resolution, orientation, and FOVs, with promising initial results demonstrated on whole-body CT images. The overall diversity of orientation and FOVs considered however appeared to be somewhat limited and simple, and additional works are needed to demonstrate if the presented model can handle more realistic variability of image resolution, orientation, and FOVs in more complex datasets.

**Questions To Address In The Rebuttal:**

The paper will be largely strengthened if more diverse settings of the FOV and orientations can be considered, or on datasets where FOV and orientations exhibit as more subtle changes of the images (rather then those shown in the whole-body CT scans).

Please add statistics of the quantitative numbers presented.

Please address the question on whether the super-resolution is limited by the extend of resolutions presented in training images.

**Special Issue:**

No

---

> ### Author Response · Authors · 2024-03-16
>
> Q1: The paper will be largely strengthened if more diverse settings of the FOV and orientations can be considered, or on datasets where FOV and orientations exhibit as more subtle changes of the images (rather than those shown in the whole-body CT scans).
>
> A1: We agree with the reviewer that a more diverse set of FOV, orientations and further use in modalities would be beneficial in the evaluation of this work. Given this was a paper based on a new methodology as opposed to an extensive validation paper, we were limited in the length of the paper and the number of results that we can generate and report.
>
> That being said we can reassure the reviewer that given the nature of the training mechanism and use of diffusion models means the model is able to generalise and extrapolate to a variety of conditionings and diverse set of image geometries. Further work we have carried out has shown the ability of the model to generate a wider range of FOV’s i.e. smaller more concentrated regions of interest like just around the bladder and prostate or just the neck region.
>
>
>
> Q2: Please add statistics of the quantitative numbers presented.
>
> A2: We agree the importance of statistical measures in the quantitative numbers presented. The paper does highlight statistically significant results in bold in the table of results (p < 0.01) however reporting the standard deviation is also important and has been added to the results displayed in the paper.
>
>
>
> Q3: Please address the question on whether the super-resolution is limited by the extend of resolutions presented in training images. - it is not clear if it requires all the resolutions to be present in the training data
>
> A3: The resolution of samples is mainly limited by the resolution seen in the training data as a floor and a ceiling. This is also because the model relies on a VQ-VAE being able to reconstruct such images at the given dimension which is limited by the resolution in the training data. Nevertheless, diffusion models have strong generalisability properties at being able to extrapolate conditioning values [1]. Any intermediate values unseen in training data can still be generated within the lower and upper bound of what is seen in the training data. As the lower and upper bound of resolutions you would want to be reconstructed would be needed to train the VQ-VAE it is of course expected the same resolution ranges would be used to train the LDM, however an interesting follow up experiment would be to narrow the range of resolutions seen by the LDM (whilst the VQ-VAE is given the full range that it can reconstruct) and seeing if it can then extrapolate beyond the seen resolutions in training. We intend to explore this in future work.
>
>
>
> [1]- Pinaya, W.H., Tudosiu, P.D., Dafflon, J., Da Costa, P.F., Fernandez, V., Nachev, P., Ourselin, S. and Cardoso, M.J., 2022, September. Brain imaging generation with latent diffusion models. In MICCAI Workshop on Deep Generative Models (pp. 117-126). Cham: Springer Nature Switzerland.

---

> > ### Comment · Reviewer_uASK · 2024-03-20
> >
> > Thanks for adding the standard deviation to the results. My other two questions were mostly responded as a future work, but seeing my fellow reviewers' comments, I'm happy to adjust my rating to a higher level.

---

### Official Review · Reviewer_mUvm · 2024-02-29

**Confidence:** 4
**Preliminary Rating:** 5
**Recommendation:** Oral
**Final Rating:** 5

**Summary:**

The author present a spatial conditional method to take control over the resolution, field of view and orientation of image generated by the latent diffusion model. Moreover, the author showcase a emergent behaviour of super-resolution by repeated diffusion and denoising process.

**Strengths:**

1. The idea of the proposed spatial condition is very interesting and easy to implement.

2. The qualitative results seem to be very promising.

3. The emergent behaviour of super-resolution is very effective.

**Weaknesses:**

1. Similar to all the image synthesis study, the evaluation of the results is relatively weak.

2.  The discussion about the super-resolution achieved by repeating the noising and denoising process seems to be insufficient. Is there any way to mathematically show why this is the case? Experimentally, is there an upper bound of the image resolution by taking more cycles of diffusion?

**Detailed Comments:**

1. Using a little bit more quantitative evaluation metrics like SSIM on the super-resolution part can probably help.

2. Add more explanation about how the super-resolution is achieved by repeating the diffusion process.

**Justification Of Final Rating:**

The author clearly explain the intuition of the recurrent diffusion approach they applied for super-resolution and its bottleneck in VQ-VAE. Although the computing resource required for training is expensive which can potentially affect the reproducibility, I think this is a very interesting work that should be shared with the MIDL cohort.

**Justification Of The Preliminary Rating:**

This is a very interesting work with decent novelty in general. The ability to control over the diffusion probabilistic model is very important. The proposed coordinate conditional term can effectively help control the image resolution, orientation and FOV of the generated image given a reference image.

**Questions To Address In The Rebuttal:**

1.  The discussion about the super-resolution achieved by repeating the noising and denoising process seems to be insufficient. Is there any way to mathematically show why this is the case? Experimentally, is there an upper bound of the image resolution by taking more cycles of diffusion?

2. This approach requires a reference image representing the full spatial range [0, 1]. In this study, this standard is given by the whole body image. Is there any other modalities / data that can potentially benefit from this method?

3. For the implementation details, how much memory will the latent diffusion model take for training? And how much time required to converge?

**Special Issue:**

Yes

---

> ### Author Response · Authors · 2024-03-16
>
> Q1: The discussion about the super-resolution achieved by repeating the noising and denoising process seems to be insufficient. Is there any way to mathematically show why this is the case? Experimentally, is there an upper bound of the image resolution by taking more cycles of diffusion?
>
> A1: Regarding why the super-resolution approach works. It was first hypothesised we would be able to do so through the description of t-values and their influence on the structure of generated images in [1]. In this paper it was suggested that higher t values were responsible for larger structures and coarse features in images, with lower t values outlining the finer details. From this we hypothesised that this would equate to the general anatomical features of the body being defined early on in the denoising phase i.e. at high t-values, with finer details coming to light from the lower t values. Drawing parallels from super-resolution methodologies, generally these coarse features and structure is present in the low resolution data, and the finer features are those that need to be super-resolved. It was thus decided that running through the denoising steps at lower t-values for a low resolution image with a predefined structure would help to define these detailed features i.e. adding resolution to a low resolution image. Conversely using higher t-values would run the risk of altering the structure of the image. Further exploration into the relation between the latent representation of the same low and high resolution image would further showcase the structural similarity in the data with lacking details and highlight why this approach works. We agree further insight into this is useful and further research is being carried out to highlight the exact relationship and reasoning for this behaviour.
>
> As for the Upper bound of the image resolution, this is more so based on the VQ-VAE capabilities. We have found that super-resolved images via the LDM method can obtain comparable results to reconstructions of the VQ-VAE at the same resolution, however the output resolution of the VQ-VAE is limited by the data used during training, as the model would struggle to reconstruct data that is at a higher resolution than that seen during training.
>
> [1] - Ho, J., Jain, A. and Abbeel, P., 2020. Denoising diffusion probabilistic models. Advances in neural information processing systems, 33, pp.6840-6851.
>
>
>
> Q2. This approach requires a reference image representing the full spatial range [0, 1]. In this study, this standard is given by the whole-body image. Is there any other modalities / data that can potentially benefit from this method?
>
> A2:The application of this study can be applied to various other modalities/data types. For example, PET can benefit from this method, especially looking at whole body PET, scans can generally have varying FOV – heart, head/neck and prostate, often at differing resolutions as well. Being able to model a particular modality at given FOVs and Resolutions can be beneficial to compliment such datasets. Furthermore, the arbitrary spatial range of [0, 1] can be representative of a different range. For example, if one was to solely work in neuroimaging one could use the [0, 1] range to represent a full head scan, where we would be able to model such scans at varying resolutions using this same methodology.
>
> Q3: For the implementation details, how much memory will the latent diffusion model take for training? And how much time required to converge?
>
> A3: For training the model was trained on 4 A100 GPUs with batch size of 6, for a total of 6000 epochs. The total training time for this was 5 days and 19hrs however it could be observed that good quality samples were present from ~ epoch 2500 with gradual improvements seen in quality over the remainder of training.

---

### Author Response · Authors · 2024-03-16

We would like to thank all the reviewers for their feedback. We are delighted to see that all reviewers find promise in the proposed spatial conditioning mechanism in addition to the model’s emergent behaviour used for super-resolution. There are however a number of common points brought to light by the reviewers that we would like to address.

Firstly, regarding a more extensive evaluation and demonstration on further datasets showcasing varying modalities, fields of view and orientations. As this paper focused on the new methodology not validation, a large focus of the paper was the implementation of the spatial conditioning for generative modelling and its further use for super-resolution. As such we were of course limited by the page restrictions to showcase a wider evaluation whilst also providing an adequate overview of our methodologies. We agree with the reviewers that a more diverse set of validation sets ranging from different modalities to varying FOVs and orientations would be of interest to include. That being said diffusion models have already shown to be used in a range of medical imaging scenarios on various modalities and as such we have no doubt that this work would be applicable to a range of modalities. Additionally conditioning in Diffusion models has been shown to be very stable in prior works [1,2] and models are very capable of generalising and extrapolating to a variety of conditioning, which in this case would showcase images of a wide range of geometries. Research is currently being carried out to showcase more of these capabilities in a wider array of scenarios to better demonstrate and validate the abilities of this methodology.

Additionally, regarding the super-resolution emergent behaviours and its limitations. Regarding the limit to the resolution achieved by this approach, generally it is seen that the main limitation to the resolution produced by the methodology is the floor/ceiling of the resolutions seen in the training data. Qualitatively it was seen that the super-resolution approach generates super-resolved samples of equal quality to that of what the VQ-VAE can reconstruct at the desired resolution. Naturally the limit of the resolution that the VQ-VAE can reconstruct is a result of the seen data during training. As such in a way the LDM itself is not limited in its performance for super-resolution and can perform on any data at any starting resolution to any end resolution provided that the VQ-VAE is able to accurately reconstruct that data.

Although further small concerns have been noted by reviewers, no more comments have been recurring among the reviewers, and as such these have been addressed in the individual replies to each reviewer.

We would once again like to thank the reviewers for their comments and for reading our paper. We look forward to addressing any further comments following the replies made.


[1]- Pinaya, W.H., Tudosiu, P.D., Dafflon, J., Da Costa, P.F., Fernandez, V., Nachev, P., Ourselin, S. and Cardoso, M.J., 2022, September. Brain imaging generation with latent diffusion models. In MICCAI Workshop on Deep Generative Models (pp. 117-126). Cham: Springer Nature Switzerland.

[2] - Rombach, R., Blattmann, A., Lorenz, D., Esser, P. and Ommer, B., 2022. High-resolution image synthesis with latent diffusion models. In Proceedings of the IEEE/CVF conference on computer vision and pattern recognition (pp. 10684-10695).

---

### Meta-Review · Area_Chair_u7mw · 2024-04-05

**Recommendation:** Accept (Poster)
**Confidence:** 5

**Metareview:**

The reviewers agreed that this paper presents an interesting approach (spatial conditioning) to enable latent diffusion models to be invariant to image resolution, orientation, and FOVs, with promising initial results demonstrated on whole-body CT images.

---

### Decision · Program_Chairs · 2024-04-05

Accept (Poster)